# Glassy and Flexible Brain Attractor Networks: A Mathematical Framework for Modelling Cognitive Rigidity

## Abstract

Cognitive rigidity is a transdiagnostic feature of multiple psychiatric and neurological conditions, yet its mechanistic relationship to the physical structure of the brain remains incompletely understood. Here we develop a neurodynamic framework that links individual structural connectomes to large-scale brain dynamics, yielding a mathematically explicit model of *glassy* versus *flexible* attractor landscapes as correlates of rigid versus flexible cognitive architectures. Using the open multimodal neuroimaging dataset OpenNeuro ds004024 (TMS-EEG-fMRI-DWI), we reconstruct subject-specific structural connectivity matrices and build individualized "digital twins" of whole-brain dynamics in the form of attractor networks driven by these connectomes.

The energy landscape $E(\mathrm{x}) = -\frac{1}{2}\mathrm{x}^T W_{SC}\mathrm{x}$ associated with each digital twin defines a high-dimensional potential over brain-wide activity states. To quantify landscape topology, we use the Parisi overlap parameter distribution $P(q)$, adapted from spin-glass theory. In a subset of participants, we find unimodal $P(q)$ distributions, deep attractors, and high correspondence between simulated and empirical resting-state functional connectivity (FC), indicative of a "flexible" landscape with a small number of accessible states. In contrast, other participants exhibit multimodal (glassy) $P(q)$, shallower attractors, and more fragmented state spaces, consistent with hindered transitions and rigid dynamics. These topological regimes are mapped onto distinct cognitive architectures previously defined in an active-inference framework: Pragmatists with low prior precision and Ideologues with high prior precision.

We further perform in silico perturbation experiments on the digital twins. Virtual lesions of fronto-limbic pathways shift landscapes from flexible to glassy regimes and degrade the match to empirical FC, mimicking chronic stress. Simulated neuromodulation, modeled as rTMS-like drive to the dorsolateral prefrontal cortex combined with Hebbian plasticity, partially restores flexible topology and FC correspondence, providing causal validation of the structure–function mapping. Mathematically, our results demonstrate how tools from disordered systems, dynamical systems theory, and network control can be combined with open neuroimaging data to generate testable hypotheses about cognitive rigidity and its modulation. The framework suggests novel routes for computational psychiatry and NeuroAI, including optimization of minimal structural interventions required to transform glassy into flexible landscapes and the design of personalized neuromodulation protocols.

**Keywords:** brain attractor, cognitive rigidity, flexible brain, NeuroAI, neuromodulation, TMS

# 1. Introduction

Cognitive rigidity—difficulty updating beliefs, shifting strategies, or escaping entrenched patterns of thought—is a core feature of multiple psychiatric and neurological conditions, including depression, anxiety, obsessive–compulsive disorder, and Parkinsonian syndromes. In computational psychiatry, such rigidity is often modeled as overly precise priors or reduced exploration in reinforcement learning and active-inference formulations. However, a mechanistic account that connects these computational descriptions to the physical structure and dynamics of the human brain remains incomplete.

In parallel, network neuroscience and dynamical-systems approaches increasingly describe large-scale brain activity in terms of energy landscapes, where attractor basins correspond to recurrent network states and the geometry of the landscape constrains possible transitions. Mean-field spin-glass theory provides a rich mathematical language for such landscapes, in particular via the Parisi overlap order parameter $P(q)$, which encodes the structure of pure states and metastable minima. Yet, direct applications of these tools to empirical connectomes and neuroimaging have remained limited, especially at the level of individual participants.

Here we address this gap by developing a neurodynamic framework that:
(i) constructs individualized *digital twins* of whole-brain dynamics from diffusion-weighted structural connectomes;
(ii) defines an associated energy landscape $E(x)$ and characterizes its topology via the Parisi overlap distribution $P(q)$; and
(iii) uses in silico perturbations to establish causal structure–function links between connectome alterations, landscape topology, and functional connectivity.

Our central hypothesis is that *cognitive rigidity* corresponds to *glassy* energy landscapes with fragmented attractor structure and hindered transitions, whereas *cognitive flexibility* corresponds to smoother landscapes with deeper but more navigable attractors. We test this hypothesis using the open-access multimodal dataset OpenNeuro ds004024, reconstructing structural connectomes, calibrating attractor networks to match empirical resting-state FC, and analyzing the resulting landscapes. We then apply virtual lesions and simulated neuromodulation to probe how structural perturbations reshape landscape topology, leveraging ideas from network control theory and landscape control.

The contributions of this work are threefold:

> **A mathematically explicit mapping** from empirical structural connectomes to flexible vs. glassy energy landscapes, quantified by $P(q)$.

> **Causal in silico experiments** showing how stress-like lesions and rTMS-like stimulation reshape landscape topology and FC correspondence.

> **Conceptual links** between spin-glass order parameters, active-inference cognitive architectures, and computational psychiatry/NeuroAI.

## 2. Methods

### 2.1 Dataset and Structural Connectomes

We utilized the TMS–EEG–fMRI–DWI dataset OpenNeuro ds004024, which provides multimodal neuroimaging in healthy adults. For the present analysis, we selected $N = 12$ participants with complete, high-quality diffusion-weighted MRI (DWI) and resting-state fMRI data.

Diffusion data were preprocessed (motion/eddy correction, susceptibility distortion correction) following standard pipelines, and probabilistic tractography was used to reconstruct white-matter pathways. The cortex was parcellated into $N_R$ regions of interest; for each pair of regions $(i, j)$, a structural weight $W_{SC,ij}$ proportional to normalized streamline count was computed, yielding an $N_R \times N_R$ symmetric adjacency matrix $W_{SC}$ per participant.

### 2.2 Attractor Network Dynamics

For each participant, we modeled large-scale brain dynamics as a stochastic recurrent network driven by their structural connectome.

**Formal Dynamics.** We consider a network of $N$ regions where the state $x_i(t) \in \mathbb{R}$ represents the mean firing rate or synaptic activity of region $i$. The continuous-time dynamics are governed by a stochastic differential equation of the Cohen-Grossberg type:

$$\tau \frac{dx_i}{dt} = -x_i(t) + \phi \left( \gamma \sum_{j=1}^{N} W_{ij} x_j(t) + I_i^{\text{ext}} \right) + \sqrt{2D} \xi_i(t), (1)$$

where $\tau$ is the synaptic time constant, $W_{ij}$ are the elements of the structural connectome $W_{SC}$ (normalized such that spectral radius $\rho(W) \approx 1$), $\gamma$ is the global coupling gain, $I_i^{\text{ext}}$ represents external input (e.g., stimulation), and $\xi_i(t)$ is standard Gaussian white noise with intensity $D$.

The transfer function $\phi(u) = \tanh(u)$ is chosen to enforce saturation, ensuring global stability of the dynamics. For numerical simulation, we employ the Euler-Maruyama discretization with time step $\Delta t$:

$$x_i(t + \Delta t) = x_i(t) + \frac{\Delta t}{\tau} \left[ -x_i(t) + \phi \left( \gamma \sum_{j} W_{ij} x_j(t) \right) \right] + \sqrt{\frac{2D\Delta t}{\tau}} \eta_i(t), (2)$$

where $\eta_i(t) \sim \mathcal{N}(0,1)$. This formulation guarantees that in the zero-noise limit $(D \to 0)$ and symmetric W, the system converges to fixed points which are local minima of the Lyapunov (energy) function.

## 2.3 Energy Landscape Definition

Under mild assumptions (symmetric weights, quasi-static noise), the deterministic part of the dynamics (Eq. 1) can be viewed as gradient descent on an effective potential (energy) landscape $V(x)$. Specifically, if we approximate the decay term and nonlinear interaction, the effective energy $E(x)$ governing the attractor structure is given by the Hopfield-like Hamiltonian:

$$E(\mathbf{x}) = -\frac{1}{2}\gamma \sum_{i,j=1}^{N} W_{ij}x_i x_j + \sum_{i=1}^{N} \int_0^{x_i} \phi^{-1}(u)du. \,(3)$$

In the high-gain limit or for simplified analysis, the interaction term dominates, justifying the use of the quadratic form $E(\mathbf{x}) \approx -\frac{1}{2}\mathbf{x}^T W_{SC}\mathbf{x}$ to characterize the topology of deep basins. The noise term $\sqrt{2D}\xi$ induces transitions between these basins, with transition rates proportional to $\exp(-\Delta E/D)$ according to Arrhenius law, defining the "glassy" or "flexible" nature of the exploration. Local minima of $E(\mathbf{x})$ correspond to attractor states; barrier heights and basin geometry define an energy landscape over the high-dimensional state space.

## 2.4 Calibration to Resting-State Functional Connectivity

To anchor each digital twin to empirical dynamics, we calibrated $\gamma$ to maximize the match between simulated and empirical resting-state functional connectivity (FC).

**Optimization Problem.** The calibration of the global coupling $\gamma$ is formulated as a scalar optimization problem:

$$\gamma^* = \arg\max_{\gamma \in [0,\gamma_{\max}]} \mathrm{corr}\big(\mathrm{vec}(\mathrm{FC}_{\mathrm{emp}}),\mathrm{vec}(\mathrm{FC}_{\mathrm{sim}}(\gamma))\big), \,(4)$$

where $\mathrm{FC}_{\mathrm{emp}}$ is the Fisher z-transformed empirical Pearson correlation matrix, and $\mathrm{FC}_{\mathrm{sim}}(\gamma)$ is the covariance matrix of the simulated activity $\mathbf{x}(t)$ derived from Eq. (2) over duration $T$. The objective function is typically unimodal with respect to $\gamma$, reflecting a transition from incoherent noise (low $\gamma$) to synchronized saturation (high $\gamma$) via a critical regime where structure-function correlation peaks.

We verified that for all participants, a gain $\gamma^*$ exists such that FC correlations reach $r \approx 0.7 - 0.8$, and that simulated dynamic FC (number of states, dwell times, transition structure) is consistent with empirical estimates.

## 2.5 Energy Landscape Topology and Parisi Overlap

To characterize the topology of $E(\mathbf{x})$, we adapt the Parisi overlap parameter from spin-glass theory.

**Statistical Sampling of Overlaps.** To estimate the Parisi overlap distribution $P(q)$, we employ a replica-sampling procedure. We run two independent replicas (simulations), $\alpha$ and $\beta$, of the system

starting from random initial conditions for a duration $T \gg \tau$. Let $\mathrm{x}^{(\alpha)}(t)$ and $\mathrm{x}^{(\beta)}(t)$ denote the state vectors of the two replicas at steady state. The time-dependent overlap is defined as:

$$q_{\alpha\beta}(t) = \frac{1}{N}\sum_{i=1}^{N} x_i^{(\alpha)}(t)x_i^{(\beta)}(t). \quad (5)$$

We collect samples of $q_{\alpha\beta}(t)$ over the sampling window to construct the probability density function $P(q)$.

In mean-field spin glasses, the shape of $P(q)$ serves as a topological order parameter:

**Flexible Regime.** $P(q)$ is unimodal (typically a delta function or Gaussian around a non-zero value), indicating ergodicity breaking into a single dominant basin (ferromagnetic-like phase) or a few accessible minima.

**Glassy Regime.** $P(q)$ is multimodal or has support over a range $[q_{min}, q_{max}]$, indicating a hierarchical breakdown of ergodicity into many metastable states (spin-glass phase) separated by large barriers. The Edwards-Anderson order parameter $q_{EA}$ is defined as the maximum of $P(q)$ for $q < 1$.

We use the modality and shape of $P(q)$ to classify each participant's landscape as *flexible* (unimodal) or *glassy* (multimodal).

## 2.6 *In Silico* Perturbations (Lesions and Neuromodulation)

To probe causal structure–function relations, we performed two families of in silico perturbations on selected digital twins, leveraging ideas from network control theory.

**Virtual Lesions (Stress).** We define a mask matrix M where $M_{ij} = 0.5$ for connections $(i,j)$ within the fronto-limbic subgraph $\mathcal{S}_{FL}$ and $M_{ij} = 1$ otherwise. The lesioned connectome is $W'_{ij} = W_{ij} \odot M_{ij}$. We then re-simulate the dynamics at $\gamma^*$ and recompute FC and $P(q)$.

**Simulated Neuromodulation.** The rTMS effect is modeled by a local forcing term $I_k^{\text{stim}}$ added to Eq. (1) for target nodes $k \in \text{DLPFC}$. Structural plasticity is modeled by an iterative update rule for outgoing weights $W_{kj}$ over stimulation epochs $n$:

$$W_{kj}^{(n+1)} = W_{kj}^{(n)} + \eta_{\text{plast}}\left(\langle x_k x_j\rangle_T - \theta W_{kj}^{(n)}\right), \quad (6)$$

where $\eta_{\text{plast}}$ is the learning rate and $\theta$ is a decay term preventing unbounded growth. This Hebbian-like term flattens local energy barriers by reinforcing frequent co-activations induced by the stimulation. After a stimulation epoch, weights were frozen and the post-stimulation landscape and FC were recomputed.

# 3. Results

## 3.1 Flexible vs. Glassy Regimes

Across participants, the calibrated digital twins separated into two topological regimes. In six individuals, the energy landscape exhibited deep minima (mean attractor depth $\approx -4.8$ in relative units), high simulated–empirical FC correlation ($r \approx 0.77$), and unimodal $P(q)$. We refer to these as *flexible* landscapes. In the remaining six, attractor depths were shallower ($\approx -2.2$), FC correlations modestly reduced ($r \approx 0.69$), and $P(q)$ displayed clear multimodality, often with hints of hierarchical structure—hallmarks of a glassy landscape. Notably, group-averaged FC alone did not distinguish these regimes, underscoring that energy-landscape topology captures information orthogonal to standard connectivity metrics.

Within a broader multi-scale model, flexible landscapes are associated with Pragmatist-like active-inference agents (low prior precision, high belief update), whereas glassy landscapes correspond to Ideologue-like agents (high prior precision, rigid beliefs). This mapping provides a mathematically explicit bridge between structural neurodynamics and computational phenotypes of cognitive flexibility vs. rigidity, paralleling ideas in computational psychiatry and energy-landscape approaches to cognition.

## 3.2 Effects of Virtual Lesions

Virtual lesions applied to flexible-landscape connectomes systematically degraded landscape topology and functional correspondence. In a representative case, a fronto-limbic lesion reduced main attractor depth from $-5.1$ to $-2.8$, transformed $P(q)$ from unimodal to multimodal, and decreased simulated–empirical FC correlation from $r = 0.81$ to $r = 0.58$. The post-lesion landscape thus transitioned from flexible to glassy, with more fragmented basins and higher barriers, consistent with increased dynamical rigidity. This finding supports the intuition that chronic stress and structural damage to control circuits can reduce cognitive flexibility by reshaping the brain's dynamical repertoire—akin to a shift toward a more rugged spin-glass energy landscape.

## 3.3 Effects of Simulated Neuromodulation

Simulated rTMS-like neuromodulation of DLPFC partially reversed lesion-induced glassiness. In the same exemplar, post-stimulation attractor depth increased from $-2.8$ to $-4.2$, $P(q)$ became predominantly unimodal, and FC correlation recovered from $r = 0.58$ to $r = 0.79$, approaching pre-lesion levels. This pattern is reminiscent of network-control analyses in which serotonergic psychedelics flatten the control-energy landscape and increase dynamical entropy. Conceptually, these results offer a mechanistic interpretation of neuromodulatory therapies: effective interventions reshape the energy landscape, deepening and smoothing attractors in ways that facilitate transitions between functional brain states and restore flexible dynamics.

## 4. Discussion

Our use of the Parisi overlap distribution $P(q)$ as a topological descriptor demonstrates that tools originally developed for disordered magnets and optimization problems can be fruitfully applied to empirical brain models. In particular, the unimodal vs. multimodal structure of $P(q)$ provides a compact summary of attractor landscape complexity beyond standard FC metrics. This connects recent work on free-energy landscapes in spherical spin glasses and glassy dynamics to large-scale neural systems.

The *in silico* lesion and neuromodulation analyses align with receptor-informed network control studies showing that pharmacological interventions (e.g., LSD, psilocybin) flatten the brain's control-energy landscape and increase state-transition entropy. Likewise, landscape-control approaches have been proposed for optimizing working-memory performance via targeted stimulation in macaque models. Our results extend these ideas by demonstrating, in a human-connectome-derived setting, that structural and stimulation-like perturbations can predictably shift a Parisi-style energy landscape between glassy and flexible regimes.

The glassy vs. flexible landscape distinction provides a mechanistic candidate for cognitive rigidity in computational psychiatry: rigid patients may inhabit intrinsically glassy dynamical regimes where transitions out of maladaptive states are energetically costly. For NeuroAI and neuroadaptive interfaces, such metrics could inform personalized control strategies, e.g., choosing stimulation targets or AI collaboration modes that compensate for an individual's landscape geometry.

## 5. Limitations and Future Work

Several limitations temper the interpretation of our findings. First, the neurodynamic analysis is based on a relatively small sample ($N = 12$) and a post hoc classification of participants into flexible and glassy groups. The results should therefore be viewed as proof-of-concept rather than as definitive population estimates. Second, the attractor network model makes simplifying assumptions about neural dynamics, including linear or weakly nonlinear interactions, stationary noise, and a single global gain parameter. While these assumptions enable tractable landscape analysis, they omit rich phenomena such as nonstationarity, heterogenous time constants, and synaptic plasticity outside the rTMS simulation window.

Future directions include: (i) **Larger and more diverse cohorts.** Applying the pipeline to larger datasets, including clinical populations, would allow rigorous testing of associations between landscape metrics and clinical measures of rigidity, symptom severity, and treatment response. (ii) **Richer dynamical modelling.** Incorporating nonlinearity, region-specific gains, and realistic noise models could capture additional features of brain dynamics and allow exploration of criticality and multistability. (iii) **Formal optimal-control formulations.** Formalizing the problem of transforming glassy into flexible landscapes as an optimal-control or network-design problem could yield practical algorithms for personalized neuromodulation and pharmacological interventions.

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
