# OpenReview forum: "Glassy and Flexible Brain Attractor Networks: A Mathematical Framework for Modelling Cognitive Rigidity"
_mathai.club/MathAI/2026/Conference — MathAI 2026 Conference Submission_

### Official Review · Reviewer_qNjq · 2026-03-12
**Neurodynamic framework linking individual structural connectomes to large-scale brain dynamics**

**Rating:** 6
**Confidence:** 4

**Review:**

Summary: The paper develops a neurodynamic framework linking individual structural connectomes to large-scale brain dynamics. It constructs personalized "digital twins" using attractor networks driven by empirical connectomes derived from the OpenNeuro ds004024 dataset. The authors characterize the resulting energy landscapes using the Parisi overlap parameter (P(q)) from spin-glass theory, classifying participants into "flexible" (unimodal P(q), deep attractors) and "glassy" (multimodal P(q), fragmented) regimes. These regimes are mapped onto cognitive architectures (Pragmatists vs. Ideologues). The paper validates the framework through in silico perturbations (virtual lesions and simulated rTMS), showing that structural damage shifts landscapes toward glassiness, while neuromodulation can restore flexibility.
Scores:
MATHEMATICAL RIGOR (7/10): The paper applies statistical physics (spin glass theory) and dynamical systems theory to neuroimaging data. The definition of the energy landscape and the use of the Parisi overlap parameter (P(q)) are mathematically grounded. The application of Cohen-Grossberg dynamics is appropriate.
NOVELTY & CONTRIBUTION (8/10): The specific application of "glassy" vs "flexible" landscapes to classify cognitive styles (Pragmatist vs. Ideologue) in a subject-specific "digital twin" framework is novel. The simulation of rTMS effects (Hebbian plasticity) to reshape the landscape is a creative theoretical contribution.
RELEVANCE TO MATHAI (9/10): Strong fit. It bridges neuroscience (Life Sciences) with advanced physics and dynamical systems mathematics, aligning well with the NeuroAI theme.
TECHNICAL QUALITY (7/10): The methodology is sound, though limited by the small sample size (N=12). The calibration of the digital twin to resting-state FC is a standard validation step. The "glassy" regime identification is theoretically sound.
CLARITY & PRESENTATION (9/10): Well-written and structured. The problem statement, methods, and results are clearly delineated. The connection between physical theory and cognitive phenotype is well-articulated.

Recommendation:
Accept. The paper offers a sophisticated mathematical perspective on a critical neuroscience problem. Despite the limited sample size, the theoretical framework is robust, novel, and highly relevant to the intersection of mathematics and AI. The "digital twin" approach combined with spin-glass theory provides a compelling new direction for computational psychiatry.

Comments to Recommendation:
Strengths: Mathematical Depth: The paper moves beyond standard correlation-based neuroscience by employing rigorous tools from dynamical systems and statistical physics (Hamiltonians, Parisi overlap).
Causal Modeling: The use of in silico perturbations (lesions and neuromodulation) allows for causal hypothesis testing within the mathematical model, a significant strength over purely observational studies.
Interdisciplinary Bridge: It successfully bridges the gap between computational psychiatry (cognitive rigidity) and mathematical physics (spin glasses), providing a quantifiable metric for cognitive flexibility.

Weaknesses:Sample Size: The study relies on a very small sample (N=12), with post-hoc classification into two groups of 6. While framed as a proof-of-concept, the statistical power to distinguish these regimes based on empirical data is limited.
Model Simplifications: The attractor network model assumes symmetric weights and a specific form of nonlinearity. While necessary for the energy function definition, these are significant simplifications of biological neural networks.

---

### Decision · Program_Chairs · 2026-03-20

**Decision:**

Accept (Oral)

**Comment:**

Dear Author(s),

On behalf of the Program Committee of the International Conference on Mathematics of Artificial Intelligence (MathAI 2026), we are pleased to inform you that your paper has been accepted for an oral presentation at MathAI 2026.

Your paper was evaluated through a rigorous two-stage review process involving both automated screening and expert review by members of the Program Committee. The reviewers recognized the quality and contribution of your work.

Presentation details:

- Format: Oral presentation (15–20 minutes + 5 minutes Q&A)
- Mode: You may present either in person (offline) at the conference venue in Sirius, Russia, or remotely via Zoom. Please indicate your preferred mode when confirming your participation.
- Conference dates: Marh 30 - April 3, 2026
- Website: https://mathai.club

Next steps:

1. Please confirm your participation and presentation mode by replying to this email mathai.club@yandex.ru no later than March 15, 2026 18:00 Moscow time.
2. If you plan to attend in person, the organizing committee will provide accommodation details separately.
3. Please prepare your final camera-ready manuscript according to the formatting guidelines available at https://mathai.club and upload it to OpenReview by March 15, 2026 18:00 Moscow time.

Should you have any questions regarding the program, logistics, or your presentation slot, please do not hesitate to contact us.

We look forward to your contribution to MathAI 2026.

With kind regards,

MathAI 2026 Program Committee
International Conference on Mathematics of Artificial Intelligence
https://mathai.club
OpenReview: https://openreview.net/group?id=mathai.club/MathAI/2026/Conference
MathAI Telegram: https://t.me/MathAI_club
IAIC International AI Committee: https://t.me/iaic_world
Email: mathai.club@yandex.ru